# Revisiting Possibilistic Fuzzy C-Means Clustering Using the Majorization-Minimization Method

**DOI:** 10.3390/e26080670

**Published:** 2024-08-06

**Authors:** Yuxue Chen, Shuisheng Zhou

**Affiliations:** School of Mathematics and Statistics, Xidian University, Xi’an 710071, China; chyuxue@hotmail.com

**Keywords:** possibilistic fuzzy c-means, fuzzy c-means, majorization-minimization, local minimum

## Abstract

Possibilistic fuzzy c-means (PFCM) clustering is a kind of hybrid clustering method based on fuzzy c-means (FCM) and possibilistic c-means (PCM), which not only has the stability of FCM but also partly inherits the robustness of PCM. However, as an extension of FCM on the objective function, PFCM tends to find a suboptimal local minimum, which affects its performance. In this paper, we rederive PFCM using the majorization-minimization (MM) method, which is a new derivation approach not seen in other studies. In addition, we propose an effective optimization method to solve the above problem, called MMPFCM. Firstly, by eliminating the variable $V\in R^{p\times c}$, the original optimization problem is transformed into a simplified model with fewer variables but a proportional term. Therefore, we introduce a new intermediate variable $s\in R^{c}$ to convert the model with the proportional term into an easily solvable equivalent form. Subsequently, we design an iterative sub-problem using the MM method. The complexity analysis indicates that MMPFCM and PFCM share the same computational complexity. However, MMPFCM requires less memory per iteration. Extensive experiments, including objective function value comparison and clustering performance comparison, demonstrate that MMPFCM converges to a better local minimum compared to PFCM.

## 1. Introduction

Cluster analysis is one of the most important topics in machine learning [1], and it has been widely applied in various fields, such as data mining [2], image processing [3], and pattern recognition [4]. Clustering algorithms are a type of unsupervised learning method, aiming to partition datasets into multiple clusters based on similarity measures, ensuring that samples within the same cluster are similar [5,6].

Generally, clustering algorithms are divided into hard and soft clustering schemes [7]. The c-means algorithm [8] is the most widely used hard clustering method due to its speed and simplicity. However, it is sensitive to initial points and often falls short of finding an optimal solution. Xu et al. [9] explored an alternative to the c-means algorithm that retains its simplicity while mitigating its tendency to become trapped in local minima, called power k-means. The optimization objective of this method is a power mean function of the distances between sample points and cluster centers, leading to its non-convex and high-dimensional characteristics, which makes it difficult to solve directly. Therefore, the majorization-minimization (MM) method is used to derive a descent scheme. In this paper, the MM method will be employed for the first time to solve the possibilistic fuzzy c-means clustering problem.

As one of the most typical soft clustering methods, fuzzy c-means (FCM) [10,11] is a fuzzy extension method of c-means, which divides *n* data points into *c* clusters by the membership grade matrix *U*. Compared to c-means, FCM achieves better clustering performance on datasets with overlapped clusters. Due to its flexibility and robustness, FCM has been favored by researchers up to now. However, it also suffers from some drawbacks. The first is that this method is sensitive to initialization and may easily become trapped in local minima [12,13]. The iteratively re-weighted algorithm for FCM (IRWFCM) [14] was proposed to find better local minima by using the iteratively re-weighted method to solve an equivalent minimization problem of the original FCM problem. FCM with the entropy regularization method (ERFCM) [15] introduces a maximum entropy term to FCM, whose purpose is trading off fuzziness and compactness. Recently, the iteratively re-weighted method was also used to optimize the equivalent problem of ERFCM in [16]. IRWFCM and IRWERFCM both have smaller objective function values and better clustering performance with fewer iterations compared to the original optimization method.

Furthermore, FCM assigns memberships to $x_{i}$ based on the relative distance from $x_{i}$ to the *c* cluster centers. In noisy data environments, this can result in noise points that are far from the *c* clusters being assigned relatively high memberships, thereby not accurately reflecting the true clustering results [17]. To overcome this shortcoming, the possibilistic c-means (PCM) method was proposed by Krishnapuram and Keller [18], which relaxes the constraint $\sum_{j=1}^{c}u_{ij}=1\;\left(i=1,\cdots ,n\right)$ so that $u_{ij}$ better reflects what we expect for the typicality of the *i*-th sample to the *j*-th cluster. However, PCM still suffers from initialization sensitivity and is prone to obtaining a coincident clustering phenomenon [19]. Therefore, Pal et al. [20] proposed the possibilistic fuzzy c-means clustering (PFCM) algorithm, which is a kind of hybrid clustering method based on PCM and FCM, to solve the coincident clustering problem of PCM. This method not only has the stability of FCM but also partly inherits the robustness of PCM, making it widely researched and applied in many practical fields [21,22].

In contrast to traditional distance-based approaches like FCM, PCM, and PFCM, the generalized entropy-based PFCM (GEPFCM) algorithm [23] was proposed for clustering noisy data. GEPFCM utilizes distance-based functions to mitigate noise contributions on the cluster centers. An evaluation based on the distance between the actual and computed cluster centers reveals that the error of GEPFCM is approximately 80% lower than that of PFCM. Moreover, PFCM encounters challenges when dealing with feature-imbalanced multidimensional data, often leading to misclassification, as it treats all features equally. Additionally, in multiclass datasets with strong noise injection, PFCM tends to produce significant deviations and overlapping centers due to difficulties in setting membership-weight parameters and the absence of between-class relationships in possibilistic memberships. To address these issues, the feature-weighted suppressed PFCM clustering (FW-S-PFCM) [24] method was introduced. This method incorporates a feature-weighted approach and a “suppressed competitive learning” strategy into the PFCM model, enhancing its capability to handle these two types of challenging data scenarios. Interested readers can find more PFCM-type algorithms in [25,26,27,28,29,30], and the relevant citations therein.

While recent advancements in PFCM methods have demonstrated enhanced performance in various aspects, such as handling complex data, distance, hyperparameters, and robustness, they do not address the problem of whether the method finds a better local optimal solution when it converges. However, as an extension of FCM on the objective function, PFCM inevitably inherits the shortcomings of FCM, such as numerous local minimums and susceptibility to falling into poor local minima. Therefore, studying how to avoid PFCM falling into poor local minima is a worthwhile endeavor. In this study, we aim to tackle this issue by first establishing an equivalent formulation of the original PFCM problem and then designing an iterative sub-problem using the MM method. In addition, we also propose an alternative derivation method for PFCM and use the MM method to prove the equivalence of these two methods. The main contributions of this paper are summarized as follows:We propose an alternative derivation method for PFCM, which begins by formulating an equivalent and simplified optimization problem, followed by solving it using the MM method. Finally, we demonstrate the equivalence of the new derivation method with PFCM.Due to the presence of a proportional term in the derived simplified optimization problem, we further transform it into an easily solvable equivalent form by introducing a new intermediate variable *s*. Then, the MM method is employed to design an iterative sub-problem. We refer to this method as MMPFCM.The complexity analysis indicates that MMPFCM and PFCM share the same computational complexity. However, MMPFCM utilizes the intermediate variable *s* of c×1 size instead of the variable *V* of c×p size to update *U* and *T*, resulting in smaller space complexity.It is theoretically proven that when the inner loop of MMPFCM is executed only once, MMPFCM degenerates to the original PFCM method.Experimental studies show that MMPFCM obtains better local minima compared to PFCM. In addition, compared with other state-of-the-art clustering methods, MMPFCM also shows its superiority.

The rest of this paper is organized as follows. Some preliminaries including notations, PFCM, and the MM method are given in Section 2. In Section 3, we rederive PFCM using the MM method, and another effective optimization method (MMPFCM) is presented in Section 4. A theoretical analysis is given in Section 5. In Section 6, the experimental results and discussions are reported. Section 7 concludes this paper.

## 2. Related Works

### 2.1. Notations

Let $X=\left[x_{1},x_{2},\cdots ,x_{n}\right]\in R^{p\times n}$ denote the data matrix, where $x_{i}\in R^{p}$ is the *i*-th sample for i=1,2,⋯,n, and $x_{i}^{T}$ denotes the transpose of vector $x_{i}$. Let $V=\left[v_{1},v_{2},\cdots ,v_{c}\right]\in R^{p\times c}$ denote the cluster center matrix, where $v_{j}\in R^{p}$ is the centroid of the cluster $C_{j}$ for j=1,2,⋯,c. The fuzzy membership matrix is denoted by $U={\left(u_{ij}\right)}_{n\times c}$, where $u_{ij}$ represents the fuzzy membership degree of the *i*-th sample to the *j*-th cluster. The possibilistic membership matrix is denoted by $T={\left(t_{ij}\right)}_{n\times c}$, where $t_{ij}$ represents the possibilistic membership degree of the *i*-th sample to the *j*-th cluster. $u_{j}$ and $t_{j}$ are the *j*-th columns of *U* and *T*, respectively. $u_{j}^{\left(k\right)}$ and $t_{j}^{\left(k\right)}$ denote $u_{j}$ and $t_{j}$ at the *k*-th iteration, respectively. In addition, *e* is a column vector with all elements equal to 1. $f\left(\theta \left|\theta_{t}\right.\right)$ represents the function *f* evaluated at θ given the condition or parameter value $\theta_{t}$.

### 2.2. Possibilistic Fuzzy C-Means Clustering

PFCM [20] is a kind of hybrid clustering method based on PCM and FCM, which contains both membership and typicality components. Its objective function is as follows:(1)$$\begin{array}{c} \begin{array}{cc} \min_{\left(U,T,V\right)}J_{PFCM}\left(U,T,V\right)= & \sum_{i=1}^{n}\sum_{j=1}^{c}\left(au_{ij}^{m}+bt_{ij}^{q}\right){∥x_{i}-v_{j}∥}_{2}^{2}+\sum_{j=1}^{c}\gamma_{j}\sum_{i=1}^{n}{\left(1-t_{ij}\right)}^{q},\\ s.t. & \sum_{j=1}^{c}u_{ij}=1,0\leq u_{ij},t_{ij}\leq 1,\left(i=1,2,\cdots ,n\right). \end{array} \end{array}$$Here, the membership weights $a\in \left(0,+\infty \right)$ and $b\in \left(0,+\infty \right)$ control the roles of the fuzzy membership degree and possibilistic membership degree. Furthermore, the exponential factors of fuzzy membership and possibilistic membership are $m\in \left(1,+\infty \right)$ and $q\in \left(1,+\infty \right)$, respectively. $\gamma_{j}$ is a penalty factor that is usually determined by [20]:$$\gamma_{j}=K\sum_{i=1}^{n}u_{ij}^{m}{∥x_{i}-v_{j}∥}^{2}/\;\sum_{i=1}^{n}u_{ij}^{m},$$
where K>0, with the most common choice being K=1.

PFCM usually updates the fuzzy membership $u_{ij}$, the possibilistic membership $t_{ij}$, and the center $v_{j}$ alternatively by
(2)$$\begin{array}{cc} u_{ij}^{\left(k+1\right)}= & \frac{{∥x_{i}-v_{j}^{\left(k\right)}∥}^{\frac{2}{1-m}}}{\sum_{l=1}^{c}{∥x_{i}-v_{l}^{\left(k\right)}∥}^{\frac{2}{1-m}}}, \end{array}$$
(3)$$\begin{array}{cc} t_{ij}^{\left(k+1\right)}= & {\left[1+{\left(\frac{b}{\gamma_{j}}{∥x_{i}-v_{j}^{\left(k\right)}∥}^{2}\right)}^{\frac{1}{q-1}}\right]}^{-1}, \end{array}$$
(4)$$\begin{array}{cc} v_{j}^{\left(k+1\right)}= & \frac{\sum_{i=1}^{n}\left[a{\left(u_{ij}^{\left(k\right)}\right)}^{m}+b{\left(t_{ij}^{\left(k\right)}\right)}^{q}\right]x_{i}}{\sum_{i=1}^{n}\left[a{\left(u_{ij}^{\left(k\right)}\right)}^{m}+b{\left(t_{ij}^{\left(k\right)}\right)}^{q}\right]}. \end{array}$$

When the membership weight *a* in the objective function of PFCM is 0, we realize PCM within the PFCM framework, and then Equations (3) and (4) are the iterative functions, respectively. When the membership weight *b* in the objective function of PFCM is 0, we realize FCM in the PFCM framework, and then Equations (Equation 2) and (4) are the corresponding iterative functions.

### 2.3. Majorization-Minimization Method

The majorization-minimization (MM) method [31] is an iterative optimization algorithm. It works by constructing a surrogate function that is simpler to minimize.

Consider the following optimization problem:$$\min_{\theta }g\left(\theta \right),\;s.t.\;\theta \in \Theta ,$$
where Θ is a nonempty closed set in $R^{n}$ and g:Θ→R is a continuous function. The MM method iteratively reduces a series of surrogate functions $f\left(\theta \left|\theta_{t}\right.\right)$ majorizing the objective function $g\left(\theta \right)$ at the current iterate $\theta_{t}$. The principle of majorization involves two key aspects: first, equality is achieved between the surrogate function and the objective function at the current iterate $\theta_{t}$, denoted as $f\left(\theta_{t}\left|\theta_{t}\right.\right)=g\left(\theta_{t}\right)$; and second, the surrogate function dominates the objective function, meaning $f\left(\theta \left|\theta_{t}\right.\right)\geq g\left(\theta \right)$ for all θ.

The update rule of the MM method is defined as follows [9]:$$\theta_{t+1}\leftarrow \arg\min_{\theta }f\left(\theta \left|\theta_{t}\right.\right),$$
which implies the descent property
$$g\left(\theta_{t+1}\right)\leq f\left(\theta_{t+1}\left|\theta_{t}\right.\right)\leq f\left(\theta_{t}\left|\theta_{t}\right.\right)=g\left(\theta_{t}\right).$$By successively minimizing these surrogate functions, we aim to converge to a local minimum of the original optimization problem. This process continues until convergence is achieved or another termination criterion is met. It finds wide applications in various fields, including machine learning [9], signal processing [32], and statistics [33].

## 3. Alternative Derivation Method for Possibilistic Fuzzy C-Means Clustering

In this section, we propose an alternative derivation method for PFCM, which interprets its iterative process using the MM method. Firstly, a simplified optimization model, which only contains the variables *U* and *T*, is obtained by eliminating *V*. Secondly, the MM method is applied to solve the new problem. Finally, we provide proof to demonstrate that this new derivation method is the same as the original PFCM method.

### 3.1. Formulation

Let $F=U^{m}$, $G=T^{q}$, and Z=aF+bG, and then according to Equation (4), we have
$$\begin{array}{c} v_{j}=\frac{\sum_{i=1}^{n}\left(af_{ij}+bg_{ij}\right)x_{i}}{\sum_{i=1}^{n}\left(af_{ij}+bg_{ij}\right)}=\frac{X\left(af_{j}+bg_{j}\right)}{{\left(af_{j}+bg_{j}\right)}^{T}e}=\frac{Xz_{j}}{z_{j}^{T}e}. \end{array}$$Substituting $v_{j}$ into Equation (Equation 1), we have
(5)$$\begin{array}{c} \begin{array}{cc} \min_{\left(U,T\right)}J_{PFCM}\left(U,T\right)= & \sum_{i=1}^{n}\sum_{j=1}^{c}z_{ij}\left(x_{i}^{T}x_{i}-2x_{i}^{T}v_{j}+v_{j}^{T}v_{j}\right)+\sum_{j=1}^{c}\gamma_{j}\sum_{i=1}^{n}{\left(1-t_{ij}\right)}^{q}\\ = & \sum_{i=1}^{n}\sum_{j=1}^{c}z_{ij}x_{i}^{T}x_{i}-2\sum_{j=1}^{c}z_{j}^{T}X^{T}v_{j}+\sum_{j=1}^{c}z_{j}^{T}ev_{j}^{T}v_{j}+\sum_{j=1}^{c}\gamma_{j}\sum_{i=1}^{n}{\left(1-t_{ij}\right)}^{q}\\ = & \sum_{i=1}^{n}\sum_{j=1}^{c}z_{ij}x_{i}^{T}x_{i}-\sum_{j=1}^{c}\frac{z_{j}^{T}X^{T}Xz_{j}}{z_{j}^{T}e}+\sum_{j=1}^{c}\gamma_{j}\sum_{i=1}^{n}{\left(1-t_{ij}\right)}^{q}. \end{array} \end{array}$$

Because the second term in problem (Equation 5) involves a proportional term, it makes problem (Equation 5) difficult to solve directly. Therefore, we introduce the MM method to optimize this equivalent optimization problem, and the next primary issue is to find a surrogate function that is easier to optimize. Considering that
(6)$$\phi \left(Z\right)=\frac{Z^{T}X^{T}XZ}{Z^{T}e}$$
is a convex function (as proven in Appendix A), then its opposite function is concave. Based on the property of concave functions, which states that the tangent line of a concave function at any point lies above the graph at that point, the following inequality holds:$$-\phi \left(z_{j}\right)\leq -\phi \left(z_{j}^{\left(k\right)}\right)-{\left(\omega_{j}^{\left(k\right)}\right)}^{T}\left(z_{j}-z_{j}^{\left(k\right)}\right),$$
where
(7)$$\omega_{j}^{\left(k\right)}=\frac{2X^{T}Xz_{j}^{\left(k\right)}}{{\left(z_{j}^{\left(k\right)}\right)}^{T}e}-\frac{{\left(z_{j}^{\left(k\right)}\right)}^{T}X^{T}Xz_{j}^{\left(k\right)}e}{{\left[{\left(z_{j}^{\left(k\right)}\right)}^{T}e\right]}^{2}}$$
is the first-order derivative function of $\phi \left(z_{j}\right)$ with respect to $z_{j}$ at the current iterate $\left(u_{j}^{\left(k\right)},t_{j}^{\left(k\right)}\right)$. Therefore, $-{\left(\omega_{j}^{\left(k\right)}\right)}^{T}z_{j}$ is further optimized as the surrogate function for $-\phi \left(z_{j}\right)$. Substituting this surrogate function into problem (Equation 5), we have
(8)$$\begin{array}{c} \begin{array}{cc} \min_{\left(U,T\right)}J_{PFCM}\left(U,T\right)= & \sum_{i=1}^{n}\sum_{j=1}^{c}z_{ij}x_{i}^{T}x_{i}-\sum_{j=1}^{c}{\left(\omega_{j}^{\left(k\right)}\right)}^{T}z_{j}+\sum_{j=1}^{c}\gamma_{j}\sum_{i=1}^{n}{\left(1-t_{ij}\right)}^{q}\\ = & \sum_{i=1}^{n}\sum_{j=1}^{c}z_{ij}x_{i}^{T}x_{i}-\sum_{i=1}^{n}\sum_{j=1}^{c}z_{ij}\omega_{ij}^{\left(k\right)}+\sum_{j=1}^{c}\gamma_{j}\sum_{i=1}^{n}{\left(1-t_{ij}\right)}^{q}\\ = & \sum_{i=1}^{n}\sum_{j=1}^{c}\left(au_{ij}^{m}+bt_{ij}^{q}\right)\left(x_{i}^{T}x_{i}-\omega_{ij}^{\left(k\right)}\right)+\sum_{j=1}^{c}\gamma_{j}\sum_{i=1}^{n}{\left(1-t_{ij}\right)}^{q}. \end{array} \end{array}$$

Further, the update formulas for the variables *U* and *T* are derived using the alternating iteration method.

### 3.2. Optimization Procedure

We optimize problem (Equation 8) with respect to one variable, with the other variables being fixed, which leads to the following two sub-problems.

Firstly, fixing the variable *T*, the *U*-update step involves minimizing problem (Equation 8) with respect to the variable *U*, which can be expressed as follows:
(9)$$\begin{array}{c} \begin{array}{cc} \min_{U} & \sum_{i=1}^{n}\sum_{j=1}^{c}au_{ij}^{m}\left(x_{i}^{T}x_{i}-\omega_{ij}^{\left(k\right)}\right),\\ s,t, & \sum_{j=1}^{c}u_{ij}=1,\;0\leq u_{ij}\leq 1,\;\left(i=1,\cdots ,n\right). \end{array} \end{array}$$We use the Lagrange multiplier method to derive the iterative function of $u_{ij}$ as follows:(10)$$u_{ij}^{\left(k+1\right)}=\frac{{\left(x_{i}^{T}x_{i}-\omega_{ij}^{\left(k\right)}\right)}^{\frac{1}{1-m}}}{\sum_{l=1}^{c}{\left(x_{i}^{T}x_{i}-\omega_{il}^{\left(k\right)}\right)}^{\frac{1}{1-m}}}.$$

Secondly, fixing the variable *U*, the *T*-update step involves minimizing problem (Equation 8) with respect to the variable *T*, which can be expressed as follows:(11)$$\min_{T}\sum_{i=1}^{n}\sum_{j=1}^{c}bt_{ij}^{q}\left(x_{i}^{T}x_{i}-\omega_{ij}^{\left(k\right)}\right)+\sum_{j=1}^{c}\gamma_{j}\sum_{i=1}^{n}{\left(1-t_{ij}\right)}^{q}.$$Taking the derivative of Equation (Equation 11) with respect to $t_{ij}$ and setting it to zero, we have
(12)$$t_{ij}^{\left(k+1\right)}=\frac{1}{1+{\left[\frac{b\left(x_{i}^{T}x_{i}-\omega_{ij}^{\left(k\right)}\right)}{\gamma_{j}}\right]}^{\frac{1}{q-1}}}.$$

Obviously, according to Equation (Equation 7), we know that $w_{j}$ is dependent on *U* and *T*. When we obtain *U* and *T* by calculating Equations (Equation 10) and (Equation 12), *U* and *T* are repeatedly utilized to update $w_{j}$ using Equation (Equation 7). This means that we can calculate *U*, *T*, and $w_{j}$ iteratively. The detailed process for solving problem (Equation 5) is summarized in Algorithm 1.
**Algorithm 1** Alternative derivation method for PFCM1:**Input** *X* and *c*.2:**Initialize** *U*, *T*, $\gamma_{j}\;\left(j=1,2,\cdots ,c\right)$, and set k=0.3:**repeat**4:    Calculate $\omega_{j}\;\left(j=1,2,\cdots ,c\right)$ by Equation (Equation 7);5:    Calculate *U* by Equation (Equation 10);6:    Calculate *T* by Equation (Equation 12);7:    k=k+1;8:**until** convergence9:**Output** *U* and *T*.

Next, we provide proof to demonstrate that this new derivation method is the same as the original PFCM method, as shown in Theorem 1.

**Theorem** **1.**
*Algorithm 1 is equivalent to the original PFCM algorithm.*


**Proof.** From Equation (Equation 7), we can further deduce the value of each item in matrix *W* as follows:
(13)$$\omega_{ij}^{\left(k\right)}=2x_{i}^{T}v_{j}^{\left(k\right)}-{\left(v_{j}^{\left(k\right)}\right)}^{T}v_{j}^{\left(k\right)},$$
and then we have
(14)$$x_{i}^{T}x_{i}-\omega_{ij}^{\left(k\right)}={∥x_{i}-v_{j}^{\left(k\right)}∥}^{2}.$$Substituting Equation (Equation 14) into Equation (Equation 10), we observe that Equation (Equation 10) is the same as Equation (Equation 2). Similarly, substituting Equation (Equation 14) into Equation (Equation 12), we observe that Equation (Equation 12) is the same as Equation (3). Therefore, based on the above analysis, Algorithm 1 is equivalent to the original PFCM algorithm, as their final iterative functions of *U* and *T* are the same.    □

To sum up, for the original PFCM problem, we provide a solution using the MM method by proving the convexity of the proportional term in problem (Equation 5). This represents a new derivation approach not found in other studies.

## 4. Majorization-Minimization Method for Possibilistic Fuzzy C-Means Clustering

In this section, another method is proposed to optimize the problem (Equation 5), called MMPFCM. Compared to PFCM, MMPFCM obtains better local minima.

### 4.1. Formulation

Here, we first present a lemma that introduces a simple optimization problem. In this lemma, a new variable *s* is introduced to eliminate the ratio term in the optimization problem, transforming the original optimization problem into an easily solvable equivalent form. The technique of introducing new variables in the lemma provides a basis for the equivalent transformation of the optimization model in this section.

**Lemma** **1.**
*For a constant μ>0, the equation $-\frac{\eta^{2}}{\mu }=\min_{s}s^{2}\mu -2s\eta$ holds, and the optimization objective reaches its minimum when $s=\frac{\eta }{\mu }$.*


According to Lemma 1, let $\mu_{j}=z_{j}^{T}e$ and $\eta_{j}=\sqrt{z_{j}^{T}X^{T}Xz_{j}}$, and we have
(15)$$s_{j}=\frac{\sqrt{z_{j}^{T}X^{T}Xz_{j}}}{z_{j}^{T}e}.$$Therefore, in order to eliminate the proportional term in problem (Equation 5), we introduce a new variable $s={\left(s_{1},s_{2},\cdots ,s_{c}\right)}^{T}$, and then problem (Equation 5) can be transformed into
(16)$$\begin{array}{c} \min_{\left(U,T,s\right)}J_{PFCM}\left(U,T,s\right)\;\;=\sum_{i=1}^{n}\sum_{j=1}^{c}z_{ij}x_{i}^{T}x_{i}+\sum_{j=1}^{c}\left[s_{j}^{2}z_{j}^{T}e-2s_{j}\sqrt{z_{j}^{T}X^{T}Xz_{j}}\right]+\sum_{j=1}^{c}\gamma_{j}\sum_{i=1}^{n}{\left(1-t_{ij}\right)}^{q}. \end{array}$$

Before solving problem (Equation 16), the following theorem provides the equivalence proof between problem (Equation 5) and problem (Equation 16).

**Theorem** **2.**
*Problem (Equation 16) is equivalent to problem (Equation 5).*


**Proof.** In order to prove that those two problems are equivalent, we substitute $s_{j}$ in Equation (Equation 15) into problem (Equation 16), and it can be immediately concluded that problem. (Equation 16) is equivalent to problem (Equation 5).    □

Problem (Equation 16) also involves three variables *U*, *T*, and *s*. However, compared to the variable *V* of p×c size in PFCM, the variable *s* in MMPFCM is a vector with *c* elements, resulting in smaller space complexity. Next, the alternating iteration method is used to solve problem (Equation 16).

### 4.2. Optimization Procedure

We optimize problem (Equation 16) with respect to one variable, with the other variables being fixed, which leads to the following three sub-problems.

Firstly, when the variables *U* and *T* are fixed, the *s*-update step is the minimization of problem (Equation 16) with respect to the variable *s*. Because the *c* components of *s* are separable, we have the following optimization problem:
(17)$$\min_{s_{j}}s_{j}^{2}z_{j}^{T}e-2s_{j}\sqrt{z_{j}^{T}X^{T}Xz_{j}}.$$Taking the derivative of Equation (Equation 17) with respect to $s_{j}$ and setting the derivative value to zero, we obtain $s_{j}$, as shown in Equation (Equation 15).

Secondly, when the variable *s* is fixed, the optimization problem involving the variables *U* and *T* can be denoted as follows:(18)$$\begin{array}{c} \min_{\left(U,T\right)}J_{PFCM}\left(U,T\right)\;\;=\sum_{i=1}^{n}\sum_{j=1}^{c}z_{ij}x_{i}^{T}x_{i}+\sum_{j=1}^{c}\left[s_{j}^{2}z_{j}^{T}e-2s_{j}\sqrt{z_{j}^{T}X^{T}Xz_{j}}\right]+\sum_{j=1}^{c}\gamma_{j}\sum_{i=1}^{n}{\left(1-t_{ij}\right)}^{q}. \end{array}$$

Since problem (Equation 18) involves an optimization objective with a square root, the derivative for direct optimization is quite complex. Therefore, we continue to rely on the principles of the MM method to search for surrogate functions as a further optimization model. It is obvious that $X^{T}X$ is a positive semidefinite matrix, so $\sqrt{z_{j}^{T}X^{T}Xz_{j}}$ is a convex function about $z_{j}$. Then, we can immediately conclude that $-\sqrt{z_{j}^{T}X^{T}Xz_{j}}$ is a concave function about $z_{j}$. Concavity supplies the linear majorization
$$-\sqrt{z_{j}^{T}X^{T}Xz_{j}}\leq -\sqrt{{\left(z_{j}^{\left(k\right)}\right)}^{T}X^{T}Xz_{j}^{\left(k\right)}}-{\left(\alpha_{j}^{\left(k\right)}\right)}^{T}\left(z_{j}-z_{j}^{\left(k\right)}\right),$$
where
(19)$$\alpha_{j}^{\left(k\right)}=\frac{X^{T}Xz_{j}^{\left(k\right)}}{\sqrt{{\left(z_{j}^{\left(k\right)}\right)}^{T}X^{T}Xz_{j}^{\left(k\right)}}}$$
is the first-order derivative function of $\sqrt{z_{j}^{T}X^{T}Xz_{j}}$ with respect to $z_{j}$ at the current iterate $\left(u_{j}^{\left(k\right)},t_{j}^{\left(k\right)}\right)$. Therefore, $-{\left(\alpha_{j}^{\left(k\right)}\right)}^{T}z_{j}$ is chosen as the surrogate function for $-\sqrt{z_{j}^{T}X^{T}Xz_{j}}$. Substituting this surrogate function into problem (Equation 18), we have
(20)$$\begin{array}{c} \begin{array}{cc} \min_{\left(U,T\right)}J_{PFCM}\left(U,T\right)= & \sum_{i=1}^{n}\sum_{j=1}^{c}\left(af_{ij}+bg_{ij}\right)x_{i}^{T}x_{i}+\sum_{j=1}^{c}\left[s_{j}^{2}{\left(af_{j}+bg_{j}\right)}^{T}e-2s_{j}{\left(\alpha_{j}^{\left(k\right)}\right)}^{T}\left(af_{j}+bg_{j}\right)\right]\\ \;\;\;\;\;\;\; & +\sum_{j=1}^{c}\gamma_{j}\sum_{i=1}^{n}{\left(1-t_{ij}\right)}^{q}. \end{array} \end{array}$$

Further, fixing the variables *T*, the *U*-update step involves minimizing problem (Equation 20) with respect to the variable *U*, which can be expressed as follows:
(21)$$\begin{array}{c} \begin{array}{cc} \; & \min_{U}\sum_{i=1}^{n}\sum_{j=1}^{c}\left(af_{ij}\right)x_{i}^{T}x_{i}+\sum_{j=1}^{c}\left[s_{j}^{2}{\left(af_{j}\right)}^{T}e-2s_{j}{\left(\alpha_{j}^{\left(k\right)}\right)}^{T}\left(af_{j}\right)\right]\\ \; & \Leftrightarrow \\ \; & \min_{U}\sum_{i=1}^{n}\sum_{j=1}^{c}\left(af_{ij}\right)x_{i}^{T}x_{i}+\sum_{i=1}^{n}\sum_{j=1}^{c}\left[s_{j}^{2}\left(af_{ij}\right)-2s_{j}\left(af_{ij}\right)\alpha_{ij}^{\left(k\right)}\right]\\ \; & \Leftrightarrow \\ \; & \min_{U}\sum_{i=1}^{n}\sum_{j=1}^{c}u_{ij}^{m}\left(ax_{i}^{T}x_{i}+as_{j}^{2}-2as_{j}\alpha_{ij}^{\left(k\right)}\right). \end{array} \end{array}$$

$u_{ij}$ is solved using the Lagrange multiplier method, and we have
(22)$$u_{ij}^{\left(k+1\right)}=d_{ij}^{\frac{1}{1-m}}/\;\sum_{l=1}^{c}d_{il}^{\frac{1}{1-m}}$$
with $d_{ij}=x_{i}^{T}x_{i}+s_{j}^{2}-2s_{j}\alpha_{ij}^{\left(k\right)}$.

Then, fixing the variables *U*, the *T*-update step involves minimizing problem (Equation 20) with respect to the variable *T*, which can be expressed as follows
(23)$$\begin{array}{c} \begin{array}{cc} \; & \min_{T}\sum_{i=1}^{n}\sum_{j=1}^{c}\left(bg_{ij}\right)x_{i}^{T}x_{i}+\sum_{j=1}^{c}\left[s_{j}^{2}{\left(bg_{j}\right)}^{T}e-2s_{j}{\left(\alpha_{j}^{\left(k\right)}\right)}^{T}\left(bg_{j}\right)\right]+\sum_{j=1}^{c}\gamma_{j}\sum_{i=1}^{n}{\left(1-t_{ij}\right)}^{q}\\ \; & \Leftrightarrow \\ \; & \min_{T}\sum_{i=1}^{n}\sum_{j=1}^{c}\left(bg_{ij}\right)x_{i}^{T}x_{i}+\sum_{i=1}^{n}\sum_{j=1}^{c}\left[s_{j}^{2}\left(bg_{ij}\right)-2s_{j}\left(bg_{ij}\right)\alpha_{ij}^{\left(k\right)}\right]+\sum_{j=1}^{c}\gamma_{j}\sum_{i=1}^{n}{\left(1-t_{ij}\right)}^{q}\\ \; & \Leftrightarrow \\ \; & \min_{T}\sum_{i=1}^{n}\sum_{j=1}^{c}t_{ij}^{q}\left(bx_{i}^{T}x_{i}+bs_{j}^{2}-2bs_{j}\alpha_{ij}^{\left(k\right)}\right)+\sum_{j=1}^{c}\gamma_{j}\sum_{i=1}^{n}{\left(1-t_{ij}\right)}^{q}. \end{array} \end{array}$$

Taking the derivative of Equation (Equation 23) with respect to $t_{ij}$ and setting it to zero, we have
(24)tijk+1=1+bdijγj11(q−1)(q−1)−1
with $d_{ij}=x_{i}^{T}x_{i}+s_{j}^{2}-2s_{j}\alpha_{ij}^{\left(k\right)}$.

From Equation (Equation 19), we find that $\alpha_{j}$ is changed by the change in *U* and *T*. So, when we calculate *U* using Equation (Equation 22) and *T* using Equation (Equation 24), $\alpha_{j}$ is updated accordingly, which means that we can calculate $\alpha_{j}$, *U*, and *T* iteratively. This process continues until convergence is achieved or another termination criterion is met.

What needs to be pointed out is that this loop is nested within the loop of the proposed method. Therefore, we refer to this loop as the inner loop of the algorithm, with its termination condition set as the maximum number of iterations of the loop being less than or equal to *K*. It should be noted that this paper selects K=5 as the reference value for the comparative experiments, as detailed in Section 6.

In summary, a new optimization method, called MMPFCM, is summarized in Algorithm 2.
**Algorithm 2** Majorization-minimization method for possibilistic fuzzy c-means clustering (MMPFCM)1:**Input** *X* and *c*.2:**Initialize** *U*, *T*, $\gamma_{j}\;\left(j=1,2,\cdots ,c\right)$, the number of times for the inner loop *K*, and set t=0.3:**repeat**4:    Calculate $s_{j}\;\left(j=1,2,\cdots ,c\right)$ by Equation (Equation 15);5:    **for** k=1:K6:        Calculate $\alpha_{j}\;\left(j=1,2,\cdots ,c\right)$ by Equation (Equation 19);7:        Calculate *U* by Equation (Equation 22);8:        Calculate *T* by Equation (Equation 24);9:    **end for**10:    t=t+1;11:**until** convergence12:**Output** *U* and *T*.

As is well known, PFCM is an extension of FCM and PCM. Therefore, MMPFCM can also be interpreted as a generalization of MMFCM and MMPCM, where MMFCM and MMPCM are two cases when b=0 or a=0 in MMPFCM, respectively. Relevant experiments regarding these two methods are provided in Appendix B.

### 4.3. An Interesting Observation

Algorithm 2 consists of two nested loops: an inner loop and an outer loop. Through analysis, we have made an interesting observation, that is, when the inner loop is executed only once, Algorithm 2 degenerates to the original PFCM method. For a detailed analysis, refer to Theorem 3.

**Theorem** **3.**
*When K=1 in Algorithm 2, Algorithm 2 is equivalent to the original PFCM method in this case.*


**Proof.** If the inner loop is executed only once, then Algorithm 2 contains one loop, and its detailed steps are as follows: calculate $s_{j}$ using Equation (Equation 15), calculate $\alpha_{j}$ using Equation (Equation 19), calculate $u_{ij}$ using Equation (Equation 22), and calculate $t_{ij}$ using Equation (Equation 24).From Equation (Equation 19), we can further deduce the value of each item in matrix *A* as follows:
(25)$$\alpha_{ij}^{\left(k\right)}=\frac{x_{i}^{T}Xz_{j}^{\left(k\right)}}{\sqrt{{\left(z_{j}^{\left(k\right)}\right)}^{T}X^{T}Xz_{j}^{\left(k\right)}}}.$$Then, substituting Equations (Equation 15) and (Equation 25) into Equation (Equation 22), we have
$$u_{ij}^{\left(k+1\right)}=\frac{{∥x_{i}-v_{j}^{\left(k\right)}∥}^{\frac{2}{1-m}}}{\sum_{l=1}^{c}{∥x_{i}-v_{l}^{\left(k\right)}∥}^{\frac{2}{1-m}}},$$
which is the same as Equation (Equation 2). Similarly, substituting Equations (Equation 15) and (Equation 25) into Equation (Equation 24), we observe that the obtained $t_{ij}^{\left(k+1\right)}$ is similar to Equation (Equation 3). Therefore, based on the above analysis, when the inner loop is executed only once in Algorithm 2, Algorithm 2 degenerates to the original PFCM method. □

In addition, Theorem 1 proves that Algorithm 1 is equivalent to the original PFCM method, so when K=1 in Algorithm 2, Algorithm 2 is also equivalent to Algorithm 1.

## 5. Theoretical Analysis

### 5.1. Convergence Analysis

If we want to prove the convergence of Algorithm 2, it is essential to initially prove the convergence of the inner loop in Algorithm 2. Therefore, let us first direct our attention to proving the convergence of its inner loop.

**Theorem** **4.**
*The inner loop in Algorithm 2 will decrease the objective value of problem (Equation 18) in each iteration until it converges.*


**Proof.** In Equation (Equation 20), let
$$f\left(Z\right)=\sum_{i=1}^{n}\sum_{j=1}^{c}z_{ij}x_{i}^{T}x_{i}+\sum_{j=1}^{c}s_{j}^{2}z_{j}^{T}e+\sum_{j=1}^{c}\gamma_{j}\sum_{i=1}^{n}{\left(1-t_{ij}\right)}^{q}.$$Then, Equation (Equation 20) becomes $f\left(Z\right)-\sum_{j=1}^{c}2s_{j}{\left(\alpha_{j}^{\left(k\right)}\right)}^{T}z_{j}$. Let $\bar{U}$ and $\bar{T}$ be the updated *U* and *T* in each interation, and then we have
(26)$$f\left(\bar{Z}\right)-\sum_{j=1}^{c}2s_{j}{\left(\alpha_{j}^{\left(k\right)}\right)}^{T}{\bar{z}}_{j}\leq f\left(Z\right)-\sum_{j=1}^{c}2s_{j}{\left(\alpha_{j}^{\left(k\right)}\right)}^{T}z_{j}.$$Furthermore, according to $\sqrt{z_{j}^{T}X^{T}Xz_{j}}$ is convex about $z_{j}$, and convexity supplies the linear majorization
$$\sqrt{{\bar{z}}_{j}^{T}X^{T}X{\bar{z}}_{j}}-\sqrt{z_{j}^{T}X^{T}Xz_{j}}\geq {\left(\alpha_{j}^{\left(k\right)}\right)}^{T}{\bar{z}}_{j}-{\left(\alpha_{j}^{\left(k\right)}\right)}^{T}z_{j},$$Further, multiplying both sides by $\sum_{j}2s_{j}$, we have
(27)$$\begin{array}{c} \sum_{j}2s_{j}{\left(\alpha_{j}^{\left(k\right)}\right)}^{T}{\bar{z}}_{j}-\sum_{j}2s_{j}\sqrt{{\bar{z}}_{j}^{T}X^{T}X{\bar{z}}_{j}}\leq \sum_{j}2s_{j}{\left(\alpha_{j}^{\left(k\right)}\right)}^{T}z_{j}-\sum_{j}2s_{j}\sqrt{z_{j}^{T}X^{T}Xz_{j}}. \end{array}$$Adding both sides of Inequality (Equation 26) and (Equation 27) yields
$$f\left(\bar{Z}\right)-\sum_{j}2s_{j}\sqrt{{\bar{z}}_{j}^{T}X^{T}X{\bar{z}}_{j}}\leq f\left(Z\right)-\sum_{j}2s_{j}\sqrt{z_{j}^{T}X^{T}Xz_{j}}.$$Therefore, the inner loop in Algorithm 2 decreases the objective value of problem (Equation 18) in each iteration until it converges. □

In conclusion, since the inner loop in Algorithm 2 converges, Algorithm 2 also converges.

### 5.2. Complexity Analysis

For the computational complexity of MMPFCM, since the multiplication operation of matrices generally requires more time than simple addition operations, we focus solely on the multiplication operation.

In Algorithm 2, for Step 4, computing $s_{j}\;\left(j=1,\cdots ,c\right)$ needs $O\left(npc+pc+nc\right)$. For Step 6, we need $O\left(2npc+pc\right)$ to calculate $\alpha_{j}\;\left(j=1,\cdots ,c\right)$. For Step 7, computing *U* needs $O\left(np+nc\right)$. For Step 8, we need $O\left(np+nc\right)$ to calculate *T*. Therefore, the total computational complexity of MMPFCM is $O\left(\left(\left(2npc+2nc+2np+pc\right)t_{1}+npc+pc+nc\right)t_{2}\right)$, where $t_{1}$ and $t_{2}$ are defined as the number of iterations of the inner loop and the outer loop in Algorithm 2, respectively.

As a consequence, MMPFCM has the same linear complexity with respect to the number of samples, i.e., $O\left(npct_{1}t_{2}\right)$. Next, we verify the effectiveness of the proposed method through experiments.

## 6. Experiments

To verify the effectiveness and clustering performance of the proposed algorithm, experimental studies are conducted on twelve real-world datasets, all selected from the UCI Machine Learning Repository (http://archive.ics.uci.edu/ml/index.php, accessed on 26 February 2024). The specifics of these datasets are outlined in Table 1.

All the experiments are run on a personal computer with an Intel Core i5-6500 processor and a maximum memory of 16 GB for all processes. The computer operates on Windows 10 with MATLAB R2017b. The convergence criterion is set as $\left|obj\left(it\right)-obj\left(it-1\right)\right|<10^{-5}$, where $obj\left(it\right)$ represents the objective function value at the end of the it-th iteration. The results are provided in tables and figures to verify the superiority of the proposed method.

### 6.1. Evaluation Metrics

To evaluate the performance of the proposed algorithm, four external metrics, including the overall F-measure for the entire dataset ($F^{*}$), Normalized Mutual Information (NMI), Adjusted Rand Index (ARI), and purity, are used to measure the agreement between the ground truth and the clustering results produced by the algorithm [14,34,35]. For all four metrics, higher scores correspond to improved clustering quality.

Metrics that do not require the labels of data are used for the performance evaluation and are called internal metrics. Two internal validity metrics, including the DBI [36] and XB [37], are selected here. It is worth noting that for both metrics, smaller values indicate better clustering performance [5].

The objective function value, time, and number of iterations are the remaining three evaluation metrics used to indicate the efficiency of the algorithms.

### 6.2. Setting of the Iterations in the Inner Loop

Since the setting of the iterations for the inner loop in Algorithm 2 is the primary issue that needs to be addressed for the execution of this method, in this subsection, we investigate the impact of different *K* on the clustering performance of the algorithm in order to select the appropriate number of iterations for the subsequent comparative experiments.

To study the effect of *K* on MMPFCM, we set the values of *K* as 1, 2, 5, 8, 10, and 30, respectively. MMPFCM is executed 10 times under random initializations, with the clarification that the initializations are the same for different values of *K*. We record the mean and standard deviation of F*, as well as the purity, across 10 experiments for different values of *K*. The results are shown in Figure 1.

Although the curves in Figure 1 exhibit minor fluctuations under different values of *K*, they generally show an upward trend. Additionally, It can be observed that the clustering performance of the algorithm shows a significant improvement when the number of iterations in the inner loop is set within the range of 5 to 10. Therefore, we can choose any number within this range to execute MMPFCM. This new update method increases the flexibility of the original model, allowing for similar fine-tuning to achieve better performance. It should be noted that for convenience, this paper selects K=5 as the reference value for all subsequent comparative experiments.

### 6.3. Comparison between PFCM and MMPFCM

Nie et al. [38] mentioned that a bad local minimum makes the objective value not small enough, which limits the algorithm’s performance. Based on this observation, the first set of experiments is carried out to evaluate the performance of MMPCM, focusing on the objective $J_{PFCM}\left(U,T,V\right)$ in Equation (Equation 1). To more intuitively compare the performance of MMPFCM and PFCM, we run both algorithms 10 times under the same random initializations and record their objective function values. Then, we calculate the difference in the objective function values, defined as $J_{MMPFCM}$—$J_{PFCM}$. Figure 2 presents a box plot of these differences across twelve real-world datasets. The green plus sign on each box plot indicates the mean of the differences. The red plus sign indicates the outliers. The red horizontal line represents the median. The length of the box represents the interquartile range (IQR), which is the range from the first quartile (Q1) to the third quartile (Q3) and shows the middle 50% of the data distribution. The whiskers extend from the box to the highest and lowest values, excluding outliers. Typically, the length of the whiskers extends up to 1.5 times the IQR from the edges of the box.

As shown in Figure 2, based on the same optimization objective function and the same random initialization conditions, the green plus signs are all below the horizontal 0 line, indicating that the mean difference in objective function values between MMPFCM and PFCM is less than 0 for these datasets. In other words, the average objective function value of MMPFCM is lower than that of PFCM. The red horizontal lines represent the median and are also below the horizontal 0 line for most datasets. This further demonstrates that the median objective function value of MMPFCM is lower than that of PFCM for most datasets, indicating better performance. Additionally, except for the COIL20 dataset, the boxes for the other datasets are below the horizontal 0 line. Since the difference is defined as $J_{MMPFCM}$—$J_{PFCM}$, the boxes being mostly below 0 indicates that the objective function value of MMPFCM is generally lower than that of PFCM. From this analysis, we can conclude that the proposed method achieves better local optimal solutions under the same initialization conditions.

The second set of experiments involves comparing the clustering performance of these two methods. Given that both the fuzzy membership *U* and the possibilistic membership *T* in PFCM can reflect the degree of membership of a point belonging to a particular cluster, we record the DBI and XB calculated by *U* and *T* for PFCM and MMPFCM in this set of experiments. The results are listed in Table 2 and Table 3, respectively.

From Table 2 and Table 3, it is evident that regardless of whether the experimental results are obtained through *U* or *T*, the DBI and XB of MMPFCM are either less than or equal to those of PFCM, indicating that the proposed method exhibits better clustering performance on these twelve datasets. Moreover, the smaller standard deviation of MMPFCM also demonstrates its stability under the same initialization conditions.

The convergence curves of PFCM and MMPFCM on twelve real-world datasets are shown in Figure 3. The figures illustrate that both algorithms exhibit monotonically decreasing objective values over time. However, MMPFCM obtains better local minima on the SCADI, COIL20, ORL, Yale64, Isolet5, and Urban datasets. Furthermore, the time taken by both methods is at the same linear scale due to their identical linear complexities.

### 6.4. Comparison between MMPFCM and Other Methods

In order to demonstrate the performance of MMPFCM more comprehensively, MMPFCM is compared with six clustering algorithms, which include the following:Fuzzy c-means (FCM) [11];Iteratively re-weighted algorithm for fuzzy c-means (IRWFCM) [14];An effective optimization method For fuzzy c-means with entropy regularization (IRWERFCM) [16];A possibilistic fuzzy c-means clustering algorithm (PFCM) [20];Generalized entropy-based possibilistic fuzzy c-means for clustering noisy data and its convergence proof (EPFCMR) [23];A feature-weighted suppressed possibilistic fuzzy c-means clustering algorithm and its application to color image segmentation (FW-S-PFCM) [24].

IRWFCM and IRWERFCM are chosen because they utilize a novel method, namely the iteratively re-weighted method, to optimize FCM-type problems, and their advantages lie in their ability to achieve better local minima with fewer iterations. EPFCMR is a generalized entropy-based PFCM, which utilizes functions of distance, not the distance itself, to decrease noise contributions on the cluster centers. FW-S-PFCM introduces a feature-weighted method and a “suppressed competitive learning” strategy into the PFCM model, resulting in improved clustering performance. Additionally, it reduces the number of iterations, sensitivity to membership weights, and initializations of PFCM.

To facilitate a comprehensive comparison between MMPFCM and other algorithms, four external metrics on twelve real-world datasets are presented in Table 4. The values are averaged over 10 trials with random initializations. The standard deviations are given after the means, and the best results are shown in bold. Note that all experimental results are calculated by the obtained *U* for different methods. In addition, the corresponding running times of the PFCM-type algorithms are illustrated in Figure 4.

From the experimental results in Table 4 and Figure 4, the following conclusions can be made:Comparing the fourth and last columns of each dataset, MMPFCM consistently outperforms PFCM across all four clustering evaluation metrics on ten datasets. In addition, MMPFCM outperforms PFCM in terms of the ARI and FM on the ORL and USPS datasets, and MMPFCM outperforms PFCM in terms of purity on the ORL dataset. These results indicate the superiority of the proposed method under the same initialization conditions.PFCM-type clustering algorithms have better clustering results than FCM-type clustering algorithms on the SCADI, balance, Yale32, Yale64, Iris, and USPS datasets. This is because PFCM-type clustering algorithms are better equipped to handle data with noise and outliers.The total running time of FW-S-PFCM is the lowest on these twelve datasets, but this is achieved under the condition of tuning more hyperparameters. The time taken by MMPFCM and PFCM is at the same linear level, which confirms that their time complexities have the same linear relationship.

## 7. Conclusions

This article mainly revisits PFCM using the MM method. By eliminating the variable *V*, we obtain a simplified model with fewer variables, and then we provide a solution using the MM method by proving the convexity of the proportional term in this model. Through analysis, the new derivation method is shown to be equivalent to the original PFCM. In addition, we introduce a new intermediate variable *s* to transform the simplified model with a proportional term into an easily solvable equivalent form. Then, we design an iterative sub-problem using the MM method. For convenience, we refer to this method as MMPFCM. The complexity analysis indicates that MMPFCM and PFCM share the same computational complexity. However, MMPFCM uses the intermediate variables *s* of c×1 size instead of the variable *V* of p×c size to update *U* and *T*, resulting in smaller space complexity. Extensive experiments have shown that MMPFCM converges to a better local minimum compared to PFCM. In addition, this new updating approach enhances the flexibility of the original model, allowing for fine-tuning the number of iterations in the inner loop to achieve better performance.

In future work, we will try to apply this strategy to other PFCM-type clustering algorithms for the purpose of obtaining a better local minimum. In addition, since this method has the same computational complexity as the original PFCM, we are also committed to researching an accelerated version of MMPFCM to reduce its running time.

## Figures and Tables

**Figure 1 entropy-26-00670-f001:**
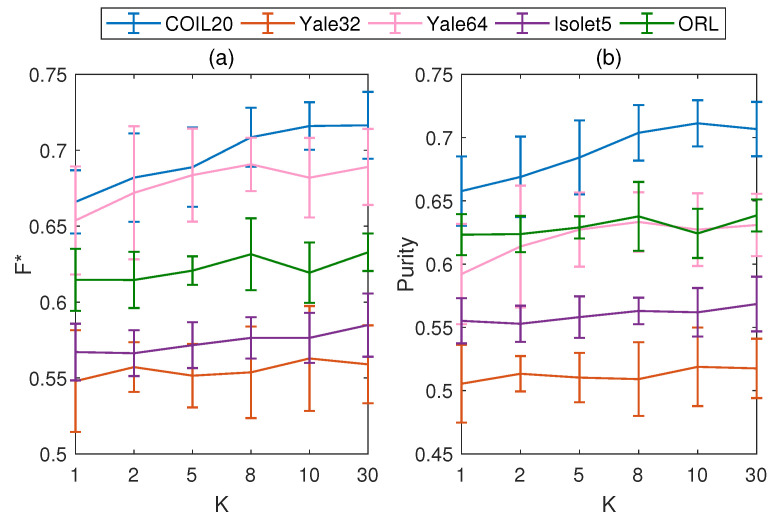
Mean and standard deviation of F*, as well as Purity, for different values of *K* on five real-world datasets, where *K* is the number of iterations in the inner loop. (**a**) Mean and standard deviation of F*. (**b**) Mean and standard deviation of Purity.

**Figure 2 entropy-26-00670-f002:**
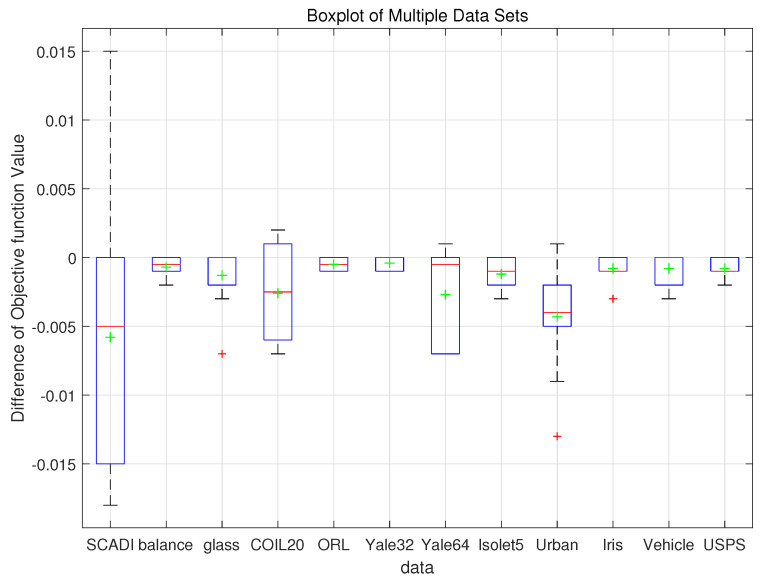
Box plot of the differences in objective function values across twelve real-world datasets under the same initialization conditions.

**Figure 3 entropy-26-00670-f003:**
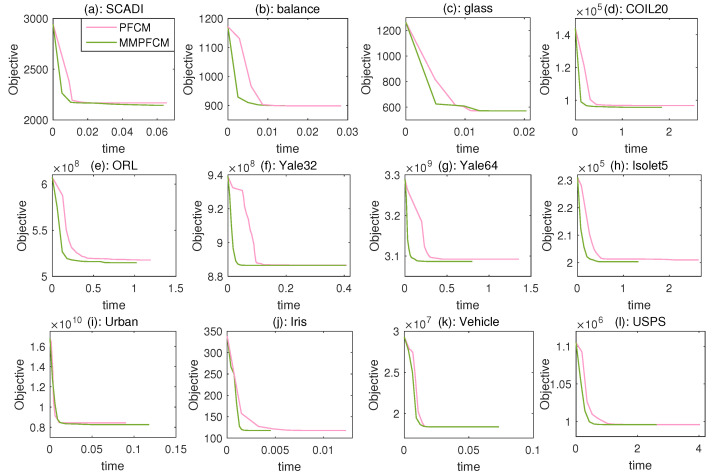
Convergence curves of PFCM and MMPFCM on twelve real-world datasets, where the two methods share the same initialization.

**Figure 4 entropy-26-00670-f004:**
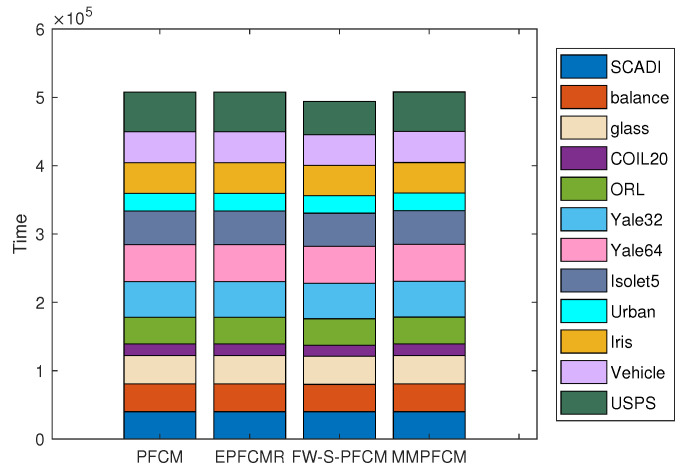
Plot of the corresponding times of the different algorithms on twelve real-world datasets.

**Table 1 entropy-26-00670-t001:** Benchmark datasets.

Datasets	Instance	Feature	Class
SCADI	70	205	7
balance	625	4	3
glass	214	9	7
COIL20	1440	1024	20
ORL	400	1024	40
Yale32	165	1024	15
Yale64	165	4096	15
Isolet5	1559	617	26
Urban	168	147	9
Iris	150	4	3
Vehicle	846	18	4
USPS	9298	256	10

**Table 2 entropy-26-00670-t002:** DBI and XB calculated by *U* for PFCM and MMPFCM on real-world datasets. The values are averaged over 10 trials with random initializations. The standard deviations are given below the means, and the best results are shown in bold.

	Datasets	SCADI	Balance	Glass	COIL20	ORL	Yale32	Yale64	Isolet5	Urban	Iris	Vehicle	USPS
		DBI											
PFCM	mean	1.430	1.611	4.427	2.284	2.045	2.598	1.860	4.471	1.558	0.639	0.682	3.367
	std	(0.137)	(0.000)	(1.856)	(0.170)	(0.058)	(0.096)	(0.127)	(0.324)	(0.129)	(0.017)	(0.007)	(0.001)
MMPFCM	mean	**1.275**	**1.610**	**4.305**	**2.258**	**2.037**	**2.561**	**1.806**	**4.448**	**1.507**	**0.634**	0.682	**3.366**
	std	(**0.097**)	(**0.000**)	(**1.707**)	(**0.119**)	(0.075)	(0.100)	(0.137)	(0.329)	(**0.078**)	(**0.017**)	(0.007)	(**0.001**)
		XB											
PFCM	mean	1.046	0.833	10.193	1.483	1.902	2.319	1.881	3.905	2.926	0.276	0.868	2.055
	std	(0.256)	(0.000)	(2.509)	(0.265)	(0.216)	(0.257)	(0.409)	(1.347)	(1.066)	(0.008)	(0.013)	(0.002)
MMPFCM	mean	**0.808**	0.833	**10.092**	**1.350**	**1.811**	**2.317**	**1.706**	**3.871**	**2.071**	**0.274**	**0.867**	**2.054**
	std	(**0.168**)	(0.000)	(**2.202**)	(**0.125**)	(**0.205**)	(0.281)	(0.429)	(**1.281**)	(**0.110**)	(**0.008**)	(**0.013**)	(**0.002**)

**Table 3 entropy-26-00670-t003:** DBI and XB calculated by *T* for PFCM and MMPFCM on real-world datasets. The values are averaged over 10 trials with random initializations. The standard deviations are given below the means, and the best results are shown in bold.

	Datasets	SCADI	Balance	Glass	COIL20	ORL	Yale32	Yale64	Isolet5	Urban	Iris	Vehicle	USPS
		DBI											
PFCM	mean	1.433	1.617	3.415	2.286	2.048	2.596	1.858	4.469	1.609	0.639	0.682	3.366
	std	(0.138)	(0.008)	(1.127)	(0.172)	(0.057)	(0.096)	(0.135)	(0.322)	(0.130)	(0.015)	(0.011)	(0.002)
MMPFCM	mean	**1.275**	**1.615**	**3.279**	**2.258**	**2.041**	**2.564**	**1.805**	**4.447**	**1.562**	**0.635**	**0.681**	**3.365**
	std	(**0.097**)	(**0.008**)	(**0.888**)	(**0.118**)	(0.077)	(0.099)	(**0.135**)	(0.329)	(**0.088**)	(0.016)	(**0.011**)	(0.003)
		XB											
PFCM	mean	0.155	0.059	3.764	1.053	0.497	0.457	0.417	1.321	1.668	0.178	0.797	0.905
	std	(0.033)	(0.000)	(0.327)	(0.172)	(0.052)	(0.050)	(0.091)	(0.451)	(0.371)	(0.005)	(0.010)	(0.001)
MMPFCM	mean	**0.125**	0.059	**3.747**	**0.973**	**0.470**	**0.454**	**0.374**	**1.309**	**1.344**	**0.177**	**0.796**	0.905
	std	(**0.022**)	(0.000)	(**0.254**)	(**0.083**)	(0.058)	(0.057)	(**0.089**)	(**0.426**)	(**0.079**)	(**0.005**)	(**0.010**)	(0.001)

**Table 4 entropy-26-00670-t004:** Experimental results of different algorithms on real-world datasets. The values are averaged over 10 trials with random initializations. The standard deviations are given after the means, and the best results are shown in bold.

Datasets	Metrics	FCM	IRWFCM	IRWERFCM	PFCM	EPFCMR	FW-S-PFCM	MMPFCM
SCADI	ARI	0.267 (0.064)	0.300 (0.001)	0.327 (0.057)	0.429 (0.064)	0.434 (0.072)	0.261 (0.042)	**0.436** (0.080)
	F*	0.535 (0.052)	0.560 (0.003)	0.581 (0.050)	0.630 (0.026)	0.632 (0.028)	0.528 (0.031)	**0.637** (0.032)
	NMI	0.492 (0.034)	0.505 (0.000)	0.518 (0.028)	0.524 (0.019)	0.523 (0.019)	0.488 (0.029)	**0.531** (0.026)
	Purity	0.734 (0.024)	**0.759** (0.005)	0.753 (0.022)	0.631 (0.009)	0.636 (0.018)	0.736 (0.024)	0.637 (0.018)
balance	ARI	0.116 (0.131)	0.114 (0.095)	0.108 (0.077)	0.128 (0.119)	0.130 (0.123)	0.130 (0.011)	**0.131** (0.125)
	F*	0.529 (0.119)	0.543 (0.078)	0.539 (0.056)	0.552 (0.089)	0.551 (0.093)	**0.577** (0.009)	0.555 (0.088)
	NMI	0.106 (0.119)	0.098 (0.086)	0.102 (0.068)	0.112 (0.100)	0.113 (0.104)	0.112 (0.009)	**0.114** (0.105)
	Purity	0.605 (0.102)	0.629 (0.067)	0.635 (0.060)	0.640 (0.075)	0.639 (0.079)	**0.656 (0.016)**	0.642 (0.077)
glass	ARI	0.235 (0.058)	0.243 (0.056)	0.241 (0.000)	0.221 (0.042)	0.175 (0.005)	0.234 (0.054)	**0.246** (0.064)
	F*	0.549 (0.017)	0.552 (0.005)	0.536 (0.000)	0.527 (0.022)	0.457 (0.015)	0.556 (0.014)	**0.558** (0.025)
	NMI	0.425 (0.032)	0.426 (0.003)	0.349 (0.000)	0.409 (0.019)	0.322 (0.005)	0.431 (0.023)	**0.435** (0.036)
	Purity	0.591 (0.009)	0.592 (0.002)	0.565 (0.000)	0.587 (0.008)	0.538 (0.017)	0.593 (0.008)	**0.594** (0.008)
COIL20	ARI	0.593 (0.026)	**0.601** (0.018)	0.590 (0.026)	0.573 (0.012)	0.575 (0.007)	0.554 (0.023)	0.593 (0.030)
	F*	0.682 (0.025)	**0.695** (0.019)	0.686 (0.018)	0.659 (0.011)	0.664 (0.006)	0.650 (0.022)	**0.695** (0.029)
	NMI	0.773 (0.012)	0.782 (0.008)	0.782 (0.015)	0.772 (0.005)	0.778 (0.005)	0.766 (0.010)	**0.783** (0.013)
	Purity	0.683 (0.030)	**0.691** (0.020)	0.679 (0.022)	0.640 (0.013)	0.640 (0.005)	0.642 (0.018)	0.689 (0.029)
ORL	ARI	0.445 (0.021)	0.459 (0.024)	0.438 (0.028)	0.456 (0.023)	0.450 (0.017)	0.427 (0.015)	**0.466 (0.013)**
	F*	0.620 (0.016)	0.630 (0.022)	0.609 (0.019)	0.622 (0.022)	0.621 (0.020)	0.603 (0.013)	**0.637 (0.010)**
	NMI	0.784 (0.006)	**0.788** (0.009)	0.776 (0.012)	0.786 (0.010)	0.785 (0.008)	0.777 (0.006)	0.784 (0.006)
	Purity	0.626 (0.020)	0.647 (0.019)	0.610 (0.019)	0.629 (0.020)	0.616 (0.017)	0.607 (0.011)	**0.650** (0.012)
Yale32	ARI	0.278 (0.011)	0.276 (0.009)	0.253 (0.029)	0.295 (0.027)	0.289 (0.023)	0.250 (0.024)	**0.304** (0.019)
	F*	0.528 (0.014)	0.524 (0.010)	0.500 (0.032)	0.537 (0.032)	0.535 (0.024)	0.496 (0.026)	**0.559** (0.025)
	NMI	0.536 (0.006)	0.538 (0.007)	0.524 (0.020)	**0.553** (0.020)	0.551 (0.019)	0.521 (0.018)	**0.553** (0.013)
	Purity	0.479 (0.023)	0.468 (0.017)	0.453 (0.023)	0.495 (0.032)	0.485 (0.022)	0.453 (0.024)	**0.518** (0.023)
Yale64	ARI	0.367 (0.023)	0.375 (0.021)	0.347 (0.043)	0.396 (0.020)	0.402 (0.025)	0.350 (0.016)	**0.410** (0.017)
	F*	0.619 (0.028)	0.635 (0.015)	0.606 (0.040)	0.654 (0.021)	0.670 (0.023)	0.602 (0.018)	**0.674** (0.019)
	NMI	0.603 (0.019)	0.615 (0.017)	0.594 (0.032)	0.623 (0.014)	0.632 (0.018)	0.599 (0.011)	**0.634 (0.011)**
	Purity	0.575 (0.021)	0.592 (0.008)	0.550 (0.033)	0.605 (0.021)	**0.619** (0.025)	0.556 (0.017)	**0.619** (0.021)
Isolet5	ARI	0.434 (0.019)	0.457 (0.023)	0.440 (0.020)	0.449 (0.019)	0.451 (0.019)	0.454 (0.014)	**0.460** (0.015)
	F*	0.559 (0.015)	0.578 (0.025)	0.552 (0.016)	0.563 (0.016)	0.570 (0.020)	0.573 (0.015)	**0.579 (0.012)**
	NMI	0.704 (0.006)	0.711 (0.010)	0.708 (0.013)	0.706 (0.008)	0.708 (0.008)	0.710 (0.013)	**0.712 (0.006)**
	Purity	0.547 (0.011)	0.565 (0.025)	0.536 (0.023)	0.536 (0.016)	0.549 (0.022)	**0.567** (0.019)	0.561 (0.013)
Urban	ARI	0.115 (0.014)	0.121 (0.005)	0.117 (0.002)	0.109 (0.003)	0.120 (0.012)	0.107 (0.030)	**0.129 (0.000)**
	F*	0.353 (0.011)	0.358 (0.002)	0.364 (0.005)	0.365 (0.006)	0.367 (0.014)	0.346 (0.033)	**0.368 (0.000)**
	NMI	0.278 (0.017)	0.286 (0.010)	0.271 (0.009)	0.273 (0.003)	0.280 (0.013)	0.266 (0.037)	**0.303 (0.000)**
	Purity	0.371 (0.013)	0.375 (0.000)	0.373 (0.000)	0.354 (0.003)	**0.380** (0.013)	0.362 (0.036)	0.369 (0.000)
Iris	ARI	0.743 (0.000)	0.743 (0.000)	0.667 (0.129)	0.780 (0.007)	0.781 (0.006)	0.716 (0.000)	**0.783** (0.005)
	F*	0.899 (0.000)	0.899 (0.000)	0.850 (0.090)	0.917 (0.003)	0.918 (0.003)	0.885 (0.000)	**0.919** (0.002)
	NMI	0.758 (0.000)	0.758 (0.000)	0.709 (0.062)	0.763 (0.006)	0.765 (0.004)	0.742 (0.000)	**0.767** (0.008)
	Purity	0.900 (0.000)	0.900 (0.000)	0.848 (0.096)	0.918 (0.003)	**0.919** (0.003)	0.887 (0.000)	**0.919** (0.002)
Vehicle	ARI	0.123 (0.000)	0.123 (0.000)	0.117 (0.022)	0.117 (0.003)	0.129 (0.004)	0.123 (0.000)	**0.141** (0.002)
	F*	0.451 (0.000)	0.451 (0.000)	0.462 (0.033)	0.461 (0.003)	0.471 (0.003)	0.455 (0.000)	**0.479** (0.002)
	NMI	0.172 (0.000)	0.172 (0.000)	0.159 (0.017)	0.152 (0.003)	0.166 (0.005)	**0.187** (0.000)	0.184 (0.003)
	Purity	0.453 (0.000)	0.453 (0.000)	0.439 (0.022)	0.437 (0.004)	0.450 (0.003)	0.453 (0.000)	**0.458** (0.002)
USPS	ARI	0.521 (0.014)	0.527 (0.010)	0.536 (0.000)	0.562 (0.005)	0.555 (0.001)	0.526 (0.000)	**0.571 (0.000)**
	F*	0.682 (0.016)	0.688 (0.012)	0.697 (0.000)	0.720 (0.004)	0.714 (0.002)	0.689 (0.000)	**0.725 (0.000)**
	NMI	0.605 (0.006)	0.607 (0.004)	**0.613 (0.000)**	0.611 (0.001)	0.612 (0.001)	0.605 (0.000)	0.610 (0.000)
	Purity	0.718 (0.017)	0.725 (0.012)	**0.734 (0.000)**	0.724 (0.017)	0.732 (0.020)	0.725 (0.000)	0.718 (0.000)

## Data Availability

The original contributions presented in the study are included in the article, further inquiries can be directed to the corresponding author.

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
