# Peer review of "Revisiting Possibilistic Fuzzy C-Means Clustering Using the Majorization-Minimization Method"

_entropy, 2024, doi:10.3390/e26080670_

Round 1

Reviewer 1 Report

Comments and Suggestions for Authors

The paper proposes an alternative derivation method for Possibilistic Fuzzy C-Means (PFCM) cluster  analysis. This new method is based on an equivalent simplified optimization problem solved through a Majorization-Minimization method. The new method denoted as MMPFCM, is proved to be equivalent to PFCM from the computational point of view but preferable because it requires less memory and detects better local minima.

The topic is interesting and the manuscript is well written. In my opinion, the paper could be suitable for publication if the following critical points are addressed. In the case a revised version of the paper will be submitted, please highlight in the text the changes with respect to the previous version.

  1.     Page 1, line 19. Replace “Clustering” with “Cluster”.

  2.     Page 1, line 29. Please, be more specific. Explain why the k-means problem is difficult to solve. What’s the difficulty?

  3.     Page 2, line 86 and further in the text. Formula (5), formula (16), and any other formula should not be cited before being presented.

  4.     Page 13, lines 338-342. By commenting the output of Table 2, you state that “…average values, standard deviations, and minimum values are all less than or equal to those of PFCM…”, hence “…MMPFCM achieves a superior local minimum compared to PFCM…”. I see that in many cases the differences of the values are very small, maybe not relevant, and in some other cases they are negative, that is the values related to MMPCFM are greater. Are you sure about your conclusion about the performance of MMPFCM compared to that of PFCM? Can you comment on it?   

Author Response

Dear Reviewer 1,

Thank you very much for your valuable comments concerning our manuscript.
The comments are very valuable and helpful for revising and improving our paper. We have carefully evaluated the critical comments and thoughtful suggestions, revised the manuscript accordingly, and responded to these suggestions point by point. All changes made to the revised paper are marked in yellow so that they may be easily identified. (See appendix for details).

Hope these will make the paper more acceptable for publication.

Reviewer 2 Report

Comments and Suggestions for Authors

The authors proposed a new PFCM approach and introduced the so-called MM derivation. Simulation results are also included for comparison with other literature. Here are some comments.
1. The main question is why the PFCM method makes sense? Usually, we consider the likelihood function to derive the objective function. The current presentation needs to be improved.
2. Lemma 1 is very obvious.
3. There is no application at all. We are not sure where to use this new method.

Author Response

Dear Reviewer #2,

Thank you very much for the advice on our manuscript. We list the response to your kindly comments following (the original comments are in blue and the replies are in black, see appendix for details). 

Reviewer 3 Report

Comments and Suggestions for Authors

Generally, the paper is well-written and can interest readers. Some problems should be addressed before its possible publication:

  1. The sentence “Furthermore, the constraints.. [17]” (l. 47-49) is unclear (“require data points to consider”?)
  2. The formulas and approaches presented in Sect. 2 should be given with the respective references (e.g., for Sect. 2.2, (1), (2)-(3), etc.). The same applies to Sect. 3 to distinguish new ideas introduced by the authors from the ones known in the literature.
  3. It will be helpful for the readers to provide a list of notation used (e.g., in the appendix or at the beginning of the paper).
  4. I can’t find information in the paper about sources/websites for the datasets presented in Table 1.

Considering the issues mentioned above, I recommend a minor revision.

Comments on the Quality of English Language

Some improvements in the language and spell-checking are necessary. There are wrongly used words or typos errors (“literatures” l. 6, “aviod” l. 79, “Given…” l. 109, “literatures” l.186, “we have the following inequality holds” p. 5, “can refer to” l. 193, “majorzation” p. 8).

Author Response

Dear Reviewer 3,

Thank you very much for your valuable comments concerning our manuscript.
The comments are very valuable and helpful for revising and improving our paper. We have carefully evaluated the critical comments and thoughtful suggestions, revised the manuscript accordingly, and responded to these suggestions point by point. All changes made to the revised paper are marked in yellow so that they may be easily identified. (See appendix for details).

Hope these will make the paper more acceptable for publication.

Round 2

Reviewer 2 Report

Comments and Suggestions for Authors

Thank you for the revised version. However, the revised version is not significantly improved.